# YouTube as a Source of Patient Information for Meningiomas: A Content Quality and Audience Engagement Analysis

**DOI:** 10.3390/healthcare10030506

**Published:** 2022-03-10

**Authors:** Michał Krakowiak, Justyna Fercho, Kaja Piwowska, Rami Yuser, Tomasz Szmuda, Paweł Słoniewski

**Affiliations:** 1Neurosurgery Department, Medical University of Gdansk, 80-952 Gdansk, Poland; justyna.fercho@gumed.edu.pl (J.F.); tomasz.szmuda@gumed.edu.pl (T.S.); pslonie@gumed.edu.pl (P.S.); 2Student’s Scientific Circle of Neurosurgery, Neurosurgery Department, Medical University of Gdansk, 80-952 Gdansk, Poland; kaja.piwowska@gumed.edu.pl (K.P.); ramiyuser@gumed.edu.pl (R.Y.)

**Keywords:** meningiomas, DISCERN scores, internet, neurosurgery, online learning, YouTube, quality

## Abstract

YouTube (YT) has become a popular health information reservoir. In this study, we aimed to evaluate the content and quality of YT videos as a source of patient information for meningiomas. A YT search was conducted for the following terms: “meningioma”, “meningiomas”, “meningeal tumor”, and “psammoma”. A total of 119 videos were examined by five independent raters, using validated quality criteria, including the Quality Criteria for Consumer Health Information (DISCERN), the Journal of the American Medical Association instrument (JAMA), and the Global Quality Score (GQS). The mean DISCERN score was 35.6 points, while the mean GQS and JAMA scores were 2.4 and 1.8, respectively. The majority of the videos were produced in the United States (37.82%). Moreover, 47.9% of the evaluated videos were graded as “poor” and only 9.24% were “good”. Statistically higher scores in all three scoring systems were associated with the following information: tumor localization, clinical manifestations, indications for surgery, treatment options, risks, adjuvant therapies, results, follow up, diagrams, and those that featured a doctor as the speaker. Misleading information was presented in 35 productions. Our findings show that the overall quality of YT on the topic of meningiomas is defective, and requires further improvement and evaluation.

## 1. Introduction

With YouTube (YT) currently being the most widespread, multinational video platform, with over two billion active monthly users of all ages, it is a simple and quick method to find information on any given topic. YT can be a learning resource for target audiences consisting of patients and their families, especially during the COVID-19 pandemic. Therefore, videos published on YT should be regularly checked, in terms of their quality, and improved, as they constitute a source of medical knowledge for a large group of recipients without medical education.

So far, the quality of videos relating to neurological and neurosurgical conditions, such as migraine, glioblastoma, stroke or brain aneurysm, has been assessed [1,2,3,4]. Only one study evaluated the quality of YT videos about meningiomas. However, the assessment was only conducted for meningioma treatment [5]. Thus, a large amount of valuable information about meningiomas still needs to be covered.

Meningiomas are the most common primary intracranial tumors in adults. Believed to be derived from meningothelial cells of arachnoid granulations of the arachnoid mater, meningiomas are slow growing and, in the majority of cases, benign central nervous system (CNS) neoplasms [6]. The symptoms of meningeal tumors depend on the localization of the lesion and the mass effect they evoke [7]. Common symptoms include headache, focal cranial nerve deficits, seizures, and cognitive change [8]. The current treatment options for meningiomas include surgery, radiotherapy, Gamma Knife surgery, and chemotherapy in selected cases.

Our aim was to evaluate the quality of YT videos about meningiomas for patients, their families, and other concerned individuals, using appropriately validated scientific scoring systems.

## 2. Materials and Methods

### 2.1. Search Strategy and Data Collection

A YT search was conducted on 10 April 2021 using a Google Chrome browser in “incognito mode”. A cleared search history, no Google account attachment, and ‘‘relevance-based ranking” was used to avoid personalization of the results. The following search terms were used for data extraction: “meningioma”, “meningiomas”, “meningeal tumor”, and “psammoma”. The first 75 findings were all included to achieve robust sampling. All researched data were publicly available, and the study did not include human or animal participation; thus, no ethics committee or YT permission was necessary.

### 2.2. Inclusion and Exclusion Criteria

With the use of the above search criteria, a total of 300 videos were found. Duplicates were removed and videos in parts were treated as one production (with views, time, and comments added together). Non-English, “pronunciation psammoma” videos and non-meningioma materials were excluded. Any material on brain tumors in general was assessed only on the part dedicated to meningiomas. 

### 2.3. Variables Extracted

The source of the upload (physician, hospital/clinic channel, health channel, or patient), video duration, and substantive content of every video were analyzed. A comparable protocol of video content was extracted (symptoms, treatment, animations, diagrams, etc.) according to previous studies [3,9,10]. In addition, the characteristic features of meningiomas (when to seek medical attention, histology, and molecular characteristics) relevant for medical and non-medical viewers were chosen and evaluated based on the rater’s experience. In order to investigate the quantitative information, the “VidIQ Vision for YouTube” plug-in was used. Audience engagement content (likes, dislikes, views, and channel subscriptions) was extracted on 11 December 2021, whereas upload dates were gathered on 27 December 2021. Two neurosurgery specialists with over 15 and 40 years of clinical experience assessed the videos to detect any substantive fallacies. Any medical information considered unproven, inaccurate, or not evidence-based was considered misleading. If any disagreements occurred between the authors, the doubts were discussed in detail. Critical errors were considered as those with potential life-threating effects.

### 2.4. Scoring System

A team of five raters—one neurosurgery specialist, a neurosurgery resident, and three medical students in their clinical years—independently evaluated the videos using the DISCERN (Quality Criteria for Consumer Health Information), GQS (Global Quality Score), and JAMA (the Journal of the American Medical Association) scoring systems [11,12,13].

The DISCERN system is a validated instrument for evaluating the reliability of health information. It consists of 16 questions on various crucial characteristics of high-quality medical publications. For each of the questions, a video can score from 1 to 5 points (with a total number of points ranging from 15 to 75). A score under 28 is classified as “very poor”, 28–38 as “poor”, 39–50 as “average”, 51–62 as ‘“good”, and 63–75 as “excellent”. Videos of “excellent” and “good” quality are considered to be a useful source of information and helpful for the patient to maintain high-quality data on the illness and treatment options; “average” videos are considered to be useful, but a patient would need an additional source of information, and “poor” and “very poor” videos are not considered to be useful and should be avoided by patients. The JAMA score ranges from 0 to 4 points, judging the following features of a publication: authorship, attribution, currency, and disclosures. The GQS is a 5-point scale (ranging from “poor” to “moderate” to “excellent”), evaluating the quality and flow of the video.

### 2.5. Audience Engagement

Introduced by Erdem and Karaca [14], specifically for evaluating the popularity of videos, video power index (VPI) and the “like ratio” are calculated according to the following formulae:(1)VPI = (likes × 100/(likes + dislikes)) × (views/day)/100;(2)Like Ratio = [likes/(likes + dislikes)] × 100.

### 2.6. Statistical Analysis

PQStat v.1.8.0 (PQStat Software, Poznań, Poland) was used for statistical analysis [15]. Data were verified with the Shapiro–Wilk test and subsequently analyzed according to the test results. The descriptive statistics included arithmetic mean, median, range, and standard deviation (std).

The intraclass correlation coefficients (ICC) were chosen to compare the reliability between raters. Based on the dataset, a two-way random-effects model for *k* multiple raters was chosen and calculated according to the guidelines presented by Koo [16]. Spearman’s rank correlation coefficient, the Kruskal–Wallis test with Dunn–Bonferroni post hoc analysis, and Mann–Whitney U test were used. Graphical illustrations were prepared using Google Sheets (Google LLC, Mountain View, CA, USA) and Microsoft 365 (Microsoft Corporation ver.18.2008, Redmond, Washington, DC, USA).

## 3. Results

After applying the exclusion criteria, 119 videos were analyzed (Figure 1).

The largest number of videos were provided by health channels (*n* = 69), followed by hospital/clinic channels (*n* = 24), physicians (*n* = 16), and patients (*n* = 10) (Figure 2). The DISCERN results were compared between video providers, showing lower DISCERN quality of health channel videos than hospital/clinic channel videos (*p* < 0.05). 

Other video providers did not show statistical differences between DSICERN results when compared (*p* > 0.05). The United States was the country that uploaded the largest number of videos (*n* = 45). Other countries of video origin were as follows: India (*n* = 15), Australia (*n* = 2), Brazil (*n* = 1), Jordan (*n* = 1), Ireland (*n* = 1), Czech Republic (*n* = 1), and the United Arab Emirates (*n* = 1). In 52 cases, the country of origin was undefinable. Videos produced in the USA had statistically higher DISCERN/GQS scores in comparison with all other videos (*p* < 0.05). The differences in JAMA scores were not statistically significant.

The video descriptive statistics are presented in Table 1. Comments were blocked in 12 videos, as were subscribers in 3 cases.

### 3.1. Video Quality Analysis

The intraclass correlation coefficients (ICC) for consistency in GQS, JAMA, and DISCERN were 0.76 (CI: 0.68–0.82), 0.75 (CI: 0.67–0.81), and 0.7 (CI: 0.68–0.71), respectively. This indicates good and moderate reliability. The two-way random-effects model for k multiple raters was used [12]. Spearman’s rank correlation coefficient was calculated for the following: DISCERN/GQS, r = 0.74 (CI: 0,64–0,81, *p* < 0.05); DISCERN/JAMA, r = 0.59 (CI: 0.46–0.7, *p* < 0.05); GQS/JAMA, r = 0.43 (CI: 0.26–0.57, *p* < 0.05). From the 119 evaluated videos, 47.9% of the videos were graded as “poor”, 21.85% achieved “average”, 21.01% were “very poor”, and only 9.24% were “good”. Compared results of video grading with the used scoring systems is presented in Figure 3.

The mean DISCERN score was 35.6 (range 19.2–62), with a median of 33.8 (±9.2). The overall mean DISCERN score for each question (Q) was 2.73. For the GQS scale, the mean score was 2.4 (range 1–4), the median 2.4, and std 0.7. For JAMA, the mean score was 1.8 (range 1–4), the median 1.5, and std 0.7. The mean DISCERN scores for each Q are presented in Figure 4.

The first two DISCERN questions (Q1 and Q2) were rated the highest (“are the aims clear?” and “does it achieve its aims?”), whereas Q12 (explanation of no treatment consequences) and Q7 (additional sources of support and information providence) achieved the lowest scores of 2.1 and 2.2, respectively.

Data considering the video content for meningiomas are presented in Figure 5.

The DISCERN score was positively correlated with the duration of the video (*p* < 0.05, r = 0.4). Views, likes, subscribers, VPI, and the like ratio did not show any correlation with the DISCERN score. 

Videos that included information regarding tumor localization, tumor clinical manifestations, indications for surgery, treatment options, risks, adjuvant therapies, results, follow up, diagrams, and those with a doctor speaking achieved statistically higher scores in all three scoring systems according to the data extracted and shown in Figure 5. The DISCERN and GQS scores were positively correlated with the inclusion of symptoms, predisposing factors, epidemiological data, clear information, and information about when to seek medical attention. Inclusion of WHO CNS, histology, and embolization techniques only influenced the DISCERN score. The JAMA scores were statistically higher for videos providing genetic predispositions for meningiomas, procedure performance materials, radiological findings, and animations, but this increase was not observed for the other scoring systems. All the data are presented in Appendix A.

Three videos on meningiomas during pregnancy had a mean DISCERN score of 46.7. The appearance of the above information did not statistically influence the scoring systems. One of these videos was classified in the top five by DISCERN, but none were in the top five VPI scores. When a patient was the speaker, this did not influence the DISCERN or GQS scores, but these videos had statistically lower JAMA scores (*p* < 0.05). 

### 3.2. Misleading Information

There were 35 videos with misleading information. Descriptive statistics for these videos are presented in Table 2. Misinformative videos did not occur statistically different in the DISCERN and GQS results, but had a statistically lower JAMA score. 

Misinformation on the origin of meningiomas was present in 11 productions. Nine productions included over- or under-estimated epidemiological data. Five videos presented misinformation on vocabulary (e.g., “metastatetic”) and WHO classification errors (e.g., WHO 4 meningioma and “WHO 1–5 grades”). One video presented an improper surgical technique. Nine cases of critical errors were detected, and are presented in Figure 6.

### 3.3. Audience Engagement

Animations were the most engaging factor, and they statistically increased views, likes, dislikes, comments, subscriptions, and VPI. Videos including advice on when to seek medical attention had a statistically higher number of likes. Productions containing misinformation resulted in a lower like ratio. The USA had a statistically higher VPI than other countries (*p* = 0.03). These data are presented in Appendix A.

### 3.4. Top five DISCERN/VPI Videos

All the data for the five videos with the highest DISCERN and VPI scores are presented in Appendix A. The top five DISCERN videos had the following traits in common: a doctor as the speaker, presentation of treatment options, epidemiological data, risk factors, and symptoms. Only certain videos presented further significant information, such as meningioma and pregnancy (one video), definition of meningioma (three videos), and WHO classification (four videos). Out of the top five VPI videos, risk factors, pregnancy and meningiomas, WHO classification, and epidemiology were not mentioned at all, while symptoms (three videos), definition (one video), treatment (four videos), and a doctor as the speaker (three videos) were found in only a few of them. Factual errors were present in three of the top 5 DISCERN videos, as well as in the top VPI video. None of the five highest VPI videos were found among the top five DISCERN scores.

## 4. Discussion

### 4.1. The Current Study

Neurosurgical topics on YT, such as hydrocephalus, glioblastoma, arteriovenous malformations, and brain aneurysms, tend to be of insufficient educational value [2,4,9,10]. This study confirms that YT material presented on the topic of meningiomas is also of low quality. The mean DISCERN score for these meningiomas videos is classified in the ”poor” bracket, where the information presented to patients seeking knowledge on the most common CNS tumors is not fully reliable. 

The outcomes are comparable to a recent study that only evaluated videos on meningioma treatment [5]. The average DISCERN scores were almost identical among the studies, showing reliability of the results in both the recent and present papers (35.6 vs. 36.4, respectively), as well as in the number of videos achieving an “average” score. There was a discrepancy in the number of “excellent” videos (4.9%) compared to none in our study [5].

The tendency to post promotional videos, rather than educational videos, is becoming a noticeable trend with arguable ethics. Orthognathic posts on YT showed 100% patient satisfaction with their treatment and outcome [17]. This was also shown by Samuel et al., who noted that 17% of the videos contained promotional and advertisement materials in his study [18], compared to 34% in another study by Ward et al. [19]. Moreover, some uploaders try to post material that is educational at first glance, yet it is of low quality, rarely revealing references, and actually serves a marketing purpose [18]. The study by Śledzińska et al. [5] pointed out that areas of uncertainty, treatment consequences, no sources of support, and information were the lowest-rated elements of videos on meningioma treatment. Our study provided comparable results, with the addition of no warnings when the condition is left ‘untreated’. Furthermore, the balance of information between the “watch and wait” strategy and operative strategy was in favor of invasive treatment, which is a problem in surgical videos. Invasive strategies seem to overcome nonsurgical treatment. Studies have shown that 40% of YT videos provided information about nonsurgical treatment options for disc herniation [20], 12% for spondylolisthesis [21], 4% for kyphosis [20], and only 3% for rotator cuffs [22]. Although the inclusion of “treatment risks” has a positive influence on the quality of YT videos [9,10], this topic is often omitted or poorly outlined. Moreover, 3.26% of tooth removal videos described contraindications, while 15.22% mentioned complications [23]. Post-operative pain was discussed in only 15% of videos on orthognathic surgery [17]. Benign prostatic hyperplasia YT videos provided insufficient information on treatment risks and uncertainties [24]. Misinformation in meningioma was present in 29% of videos, whereas errors varied from 5% up to 77% for other videos, depending on the topic [10,25,26,27]. In 7% of videos, the presented errors could have potentially life-threatening consequences. Singh et al. showed that, in their series, 19% of videos undermined the credibility of evidence-based treatment [25]. Nonfactual statements were discovered in spinal cord stimulation videos that claimed inaccurate risks and benefits, s along with an inaccurate physiological description of the process itself [28]. Misinformative material regarding meningioma only negatively influenced the JAMA scoring system, indicating the low quality of the editorial content. The spread of misinformation is a dangerous phenomenon, considering that useless videos tend to be more engaging and popular than useful videos [29,30]. Aside from errors, these videos tend to be incomplete, since other important aspects are rarely discussed on screen. In almost 86% of cases, YT videos on meningiomas did not include a definition, and only 12 videos informed the viewer when to seek medical attention. Medical conditions during childbearing represent a topic that is often omitted; in this case, only three YT videos discussed meningiomas in pregnancy, compared to four for CO intoxication videos [31] and one for arteriovenous malformation [10]. YT is more frequently used by medical students for learning purposes [32]. With the use of meningioma videos as a learning tool, the viewer will only encounter a minority of videos with information on epidemiology and WHO classification. It is worth mentioning that in 2021, WHO published their 5th classification of tumors of the central nervous system, with changes in meningioma classification, and the European Association of Neuro-Oncology released its latest guidelines on meningioma diagnosis and management [33,34]. Moreover, the viewer is susceptible to inaccurate data on the origin of meningiomas, as this was the most common error encountered in our study.

### 4.2. Future Directions

Our results should be considered as a guideline for future video providers on what essential information needs to be included in the material. Furthermore, broader verification should be performed, in order to identify when promotional material is disguised as “educational”. The amount of misinformation suggests the need for a comparable study in the next few years, to verify whether these errors have been corrected. Future studies should evaluate how YT viewers react to foul videos, and if this has a negative impact on the doctor–patient relationship.

### 4.3. Limitations

Our analysis only included English videos, with only the first 75 results for every search term. Moreover, in more than 50% of videos, the country of origin was undefinable, due to a lack of data on the video provider channel. Although many studies have tried to evaluate YT medical material, discrepancies in evaluation methods still exist [35]. Browser setting changes could affect the video search results. There is no consensus as to whether the accepted scoring systems should be used for YT materials, as they were not primarily designed for YT video assessment [36].

## 5. Conclusions

Our study results indicate that the overall quality of YT on the topic of meningiomas is insufficient and requires error correction. Much of the current information is misleading, and does not provide a sufficient amount of information for a non-medical viewer to fully understand the most common primary CNS tumors.

## Figures and Tables

**Figure 1 healthcare-10-00506-f001:**
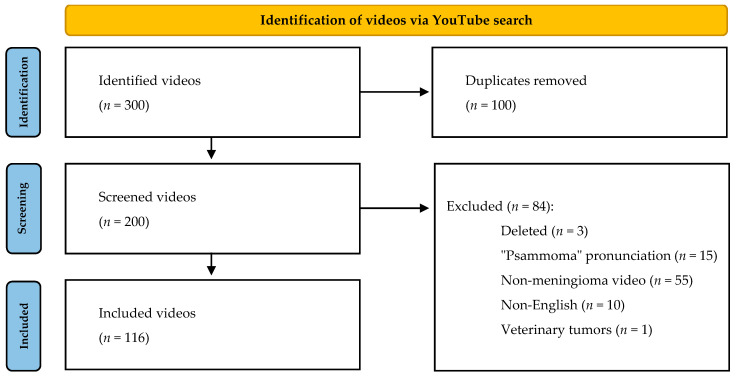
Video study inclusion.

**Figure 2 healthcare-10-00506-f002:**
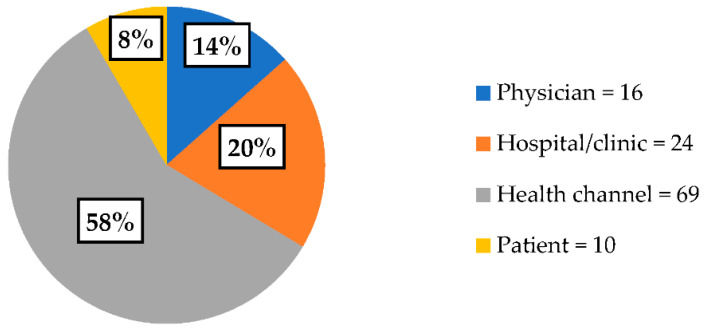
Source of upload for videos on meningiomas.

**Figure 3 healthcare-10-00506-f003:**
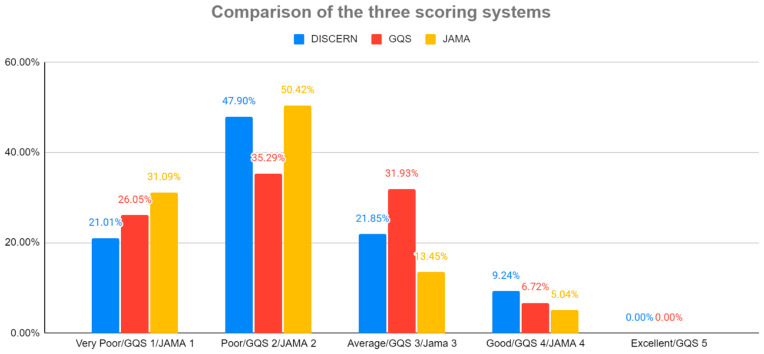
Percentage of video grades according to DISCERN, GQS, and JAMA.

**Figure 4 healthcare-10-00506-f004:**
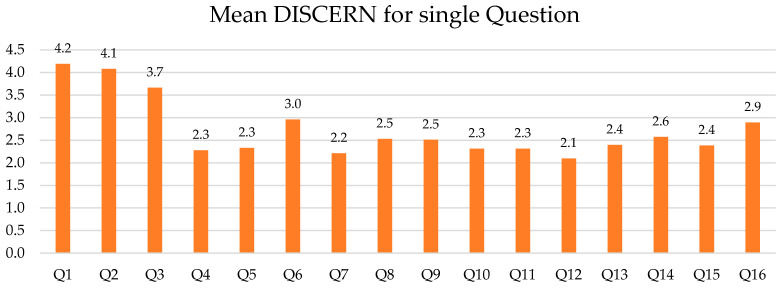
Mean score for each DISCERN question.

**Figure 5 healthcare-10-00506-f005:**
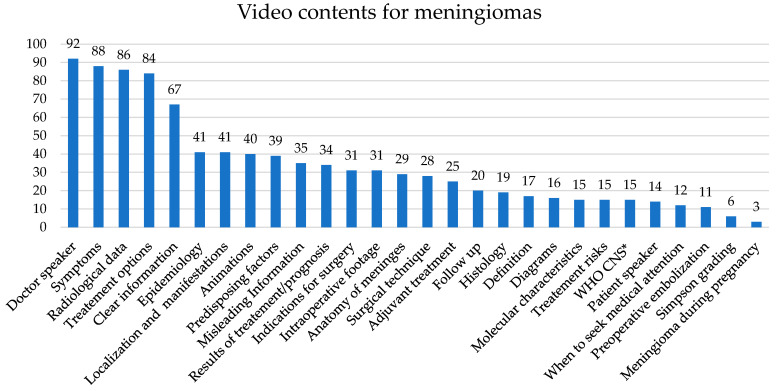
Video content for meningiomas. * WHO CNS—WHO classification of tumors of the central nervous system.

**Figure 6 healthcare-10-00506-f006:**
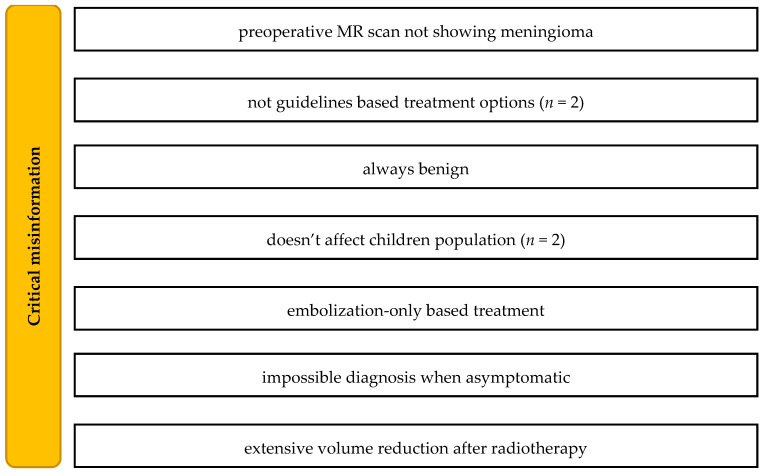
Critical misinformation in meningioma videos.

**Table 1 healthcare-10-00506-t001:** Video descriptive statistics.

	Mean	Median	Sid	Min	Max
Views	24,210.47	5027.00	68,137.98	10.00	504,633.00
Likes	224.59	50.00	659.30	0.00	6000.00
Dislikes	8.29	1.00	24.91	0.00	228.00
Comments	26.74	4.00	96.68	0.00	881.00
Subscribers	53,936.77	24,000.00	88,249.26	118.00	426,000.00
Time (s)	1133.62	506.00	1416.63	40.00	6655.00
Days since upload	1688.71	1432.00	1098.97	280.00	5347.00
VPI	16.34	3.71	55.08	0.02	510.63
Like ratio	89.52	96.55	24.58	0.00	100.00

**Table 2 healthcare-10-00506-t002:** Descriptive statistics for videos with misleading information.

	DISCERN	Views	Likes	Dislikes	Comments	Subscribers	Time (s)
Mean	34.67	30,082.29	206.40	8.49	28.68	57,136.00	1016.91
Median	30.60	4159.00	36.00	1.00	7.00	26,300.00	316.00
Standard deviation	11.08	86,380.08	429.58	21.65	76.64	85,988.86	1334.72
Minimum	19.20	10.00	0.00	0.00	0.00	129.00	51.00
Maximum	62.00	504,633.00	2200.00	125.00	419.00	424,000.00	4702.00

## Data Availability

Not applicable.

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
