# Peer review of "YouTube as a Source of Patient Information for Meningiomas: A Content Quality and Audience Engagement Analysis"

_healthcare, 2022, doi:10.3390/healthcare10030506_

Round 1
Reviewer 1 Report
The paper reports a study on the content and quality of YouTube videos on meningiomas. The topic is important, its conclusion about the low quality of these widely – accessed materials may spur further debates that could alleviate to a certain extent the negative effects of audience exposure to them, the analyses take into account multiple indicators of the video content, evaluation and audience engagement. I think there are only certain minor issues to be addressed, which would increase the readability of the paper and provide some further necessary information.
Line 96 the relevance of the Video Power Index (VPI) and the “Like Ratio” needs to be explained and supported by the previous studies that introduced them.
Line 114 “Health channels presented lower quality videos than hospital/clinic channels (p<0.05). ><0.05)” - more details are needed concerning these comparisons; for instance, on which criteria were they performed? What are the statistic indicators that emerged (e.g., means)?
Line 150 “Views, likes, subscribers VPI, and like ratio did not occur statistical significance” – in which analyses?. A similar issue is on line 162 “Three videos presenting meningiomas during pregnancy had a mean of DISCERN 46.7 but did not achieve statistical significance.”
Line 152 “Videos that included following elements: tumor localization, tumor clinical manifestations, indications for surgery, treatment options, risks, adjuvant therapies, results, 153 follow up, diagrams, and those with a doctor speaking shown statistically higher scores 154 in all 3 scoring systems” – higher than which categories?
- the development / source for the grid of the categories of video content (listed in Figure 4 and commented thereafter) needs to be explained or specified (this would argue why would these categories be relevant for this type of video material, and not others).
Author Response
Dear Reviewer,
Thank you for giving us the opportunity to submit a revised draft of our manuscript titled ”YouTube as a source of patient information for meningiomas: A content-quality and audience engagement analysis” to Healthcare. We appreciate the time and effort that you have dedicated to providing your valuable feedback on our manuscript. We are grateful for the insightful comments on our paper. We have been able to incorporate changes to reflect most of the suggestions provided.
Video Power Index (VPI) and the “Like Ratio” - explanation and reference has been added.
Line 114 - data was added
Line 150 “Views, likes, subscribers VPI, and like ratio did not occur statistical significance” – in which analyses?. information has been added
Line 152 “Videos that included following elements: tumor localization, tumor clinical manifestations, indications for surgery, treatment options, risks, adjuvant therapies, results, - information has been added
- the development / source for the grid of the categories - explanation added in section 2.3
Hopefully the implied improvements will meet the manuscript acceptance requirements.
Kind Regards,
Author
Reviewer 2 Report
The authors present meningiomas content quality analysis from the data extracted from YouTube (YT). There are serious flaws in the design of the experiment. It is not clear what, why and how videos where choosen and why YouTube only given many other streaming services. YT is a video streaming serivce which is not designed for the purpose the authors are looking into. Selection of keywords is also questionable. It is unclear what are the main contributions of this article. The inference giving in the abstract: "the overall quality of YT on the topic of meningiomas is defective and requires further improvement and evaluation" is meaningless.
For this reviewer, the paper lack any novelity, there are serious issues in the design and data collection and hence all of the evalution is inaccurate. Hence, it is not appropriate for publication in its current state. The authors should review the latest relevant literature in detail in order to understand how to improve the experiment design and how to do general/specific inference.
Author Response
Dear Reviewer,
Thank you for giving us the opportunity to submit a revised draft of our manuscript titled ”YouTube as a source of patient information for meningiomas: A content-quality and audience engagement analysis”to Healthcare. We appreciate the time and effort that you and the reviewers have dedicated to providing your valuable feedback on our manuscript. We are grateful to the reviewers for their insightful comments on our paper. We have been able to incorporate changes to reflect most of the suggestions provided by the reviewers and highlighted the changes within the manuscript.
Dear Reviewer, You have raised an important point here. However, according to Pubmed, in 2021 over 100 studies evaluated infomation posted on YT on various medical topics. In our opinion, YT as a platform became one of the main source of patient and medical staff educational tool nowadays, because it is a free, very widespread and easily accessible platform.
1) YouTube is the Most Frequently Used Educational Video Source for Surgical Preparation - doi: 10.1016/j.jsurg.2016.04.024. Epub 2016 Jun 14.
2) Using YouTube to Learn Anatomy: Perspectives of Jordanian Medical Students - doi: 10.1155/2020/6861416. eCollection 2020.
The manuscript was designed according to previous studies published in various IF Journals. Moreover, articles cited in our manuscript are from the 2022 and 2021 discussing with the latest works on this topic.
The keywords were chosen not by accident and correspond to similar works on this subject. We made the choice based on defining the recipients of the text, analyzing previous works on this topic, enabling effective long tail keyword, and we verified our selection using commonly available keyword searching tools.
We kindly ask to reconsider Your opinion on this manuscript. We look forward to hearing from you and respond to any further questions and comments you may have.
Kind regards,
Author
Reviewer 3 Report
Dear Authors; This is successful study exploring the quality of YT videos regarding the Meningioma content. I observe some issues in the content listed below needed to be addressed. Regards.
P.S.
[1] Writing
1-1 "4.Discussion" needs a new section at the start: "4.1. The current study"; "4.2. Future Directions"; "4.3. Limitations". You may renumber the subsections to make this part coherent.
1-2 Please add a list of used abbreviations in the manuscript right before the reference section. The manuscript needs to be "self-sufficient" for the readers. Example.
Abbreviations
JAMA Journal 14 of the American Medical Association instrument
[2] Statistical
2-1 A major concern here is "the potential systematic bias" in data collection for the YT videos on the Meningioma. YT policy on uploading the high quality videos may prohibit uploading them only by applying the copyright issue. This may contribute to uploading those YT videos with lower quality. Then, from statistical and computer science point of view, analyzing such poor gathered data from YT is called "garbage in-garbage out(GIGO)". This needs to be added to the limitations of the study. Please, see here for the general reference: https://en.wikipedia.org/wiki/Garbage_in,_garbage_out
My verdict here gets more strength when in line 121 the authors mention "In 52 cases, the country of origin was undefinable." This 52/119 = 43.6% of the entire dataset . This is a big problem ......
2-2 In line 101 "PQStat v.1.8.0 (PQStat Software, Poznań, Poland)" please add the related reference in the reference section.
2-3 In line 128 "Intraclass Correlation Coefficients (ICC)". How did you calculate this ? Please add its related formulae in the section "2.6. Statistical analysis".
2-4 In line 130 "The two-way random-effects model for k multiple raters was used" where is the model ? Please add its related model formulae in the section "2.6. Statistical analysis".
2-5 [Important] One figure is missing from the manuscript and that comparing the average ratings over the three rating methods in one shot. Please add this Figure in line 135.
X axis: Five ticks "very poor", "poor", "average" , "good" , "very good". Each thick has four bars together named "DISCERN", "JAMA", "CSQ", "Overall".
Y axis: percentage
The authors have already the statistics for the case of "overall" in the text. They need to add others too.
See here for an example for the above Figure:
https://stats.stackexchange.com/questions/3842/how-to-create-a-barplot-diagram-where-bars-are-side-by-side-in-r
Author Response
Dear Reviewer,
Thank you for giving us the opportunity to submit a revised draft of our manuscript titled” YouTube as a source of patient information for meningiomas: A content-quality and audience engagement analysis” to Healthcare. We appreciate the time and effort that you have dedicated to providing your valuable feedback on our manuscript. We are grateful for the insightful comments on our paper. We have been able to incorporate changes to reflect the suggestions provided.
[1] Writing - corrected
[2] Statistical
"Garbage in-garbage out" is indeed a concern when evaluating material posted without any substantive control. However, not all productions on YT are worthless. For instance, Neurosurgical Focus posts materials on YT, Dr. Aaron Cohen-Gadol shares fantastic videos on various neurosurgical topics, Seattle Science Foundation channel educates medical and the non-medical audience.
Furthermore, even surgeons prepare for surgery using YT.
1)YouTube is the Most Frequently Used Educational Video Source for Surgical Preparation - doi: 10.1016/j.jsurg.2016.04.024. Epub 2016 Jun 14.
Indeed, the amount of weak material is greater, what was pointed out in the results. That is why this manuscript covers the most common primary tumor and evaluates misinformation included in the posted movies.
2-2, 2-3, 2-4 - information and required references have been added.
2-5 [Important] - Figure added
Hopefully, the implied improvements will meet the manuscript acceptance requirements.
Kind Regards,
Author
Round 2
Reviewer 2 Report
Unfortunately, the authors haven't made any changes or updated the manuscript to improve it qualitatively and academically. Regretfully, I still cannot find any novelty or contributions of the proposed work and in my opinion, the manuscript requires extensive work in terms of experiment design, data collection, data analysis, and inference. The article in its current state is not feasible for publication.
Reviewer 3 Report
Dear Authors; most of my concerns were addressed satisfactorily. Regards.